# Chronic Inflammation in Obesity and Cancer Cachexia

**DOI:** 10.3390/jcm11082191

**Published:** 2022-04-14

**Authors:** Rosa Divella, Gennaro Gadaleta Caldarola, Antonio Mazzocca

**Affiliations:** 1ASD Nordic Walking Apulia Lifestyle, Corso Giuseppe Di Vittorio 14, 70024 Gravina in Puglia, Italy; 2Medical Oncology Unit, “Mons. R. Dimiccoli” Hospital, Viale Ippocrate 15, 70051 Barletta, Italy; gergad@libero.it; 3Interdisciplinary Department of Medicine, University of Bari School of Medicine, Piazza G. Cesare, 11, 70124 Bari, Italy

**Keywords:** chronic inflammation, obesity, cachexia

## Abstract

Chronic inflammation has long been linked to obesity and related conditions such as type 2 diabetes and metabolic syndrome. According to current research, the increased risk of cancer in people with certain metabolic diseases may be due to chronic inflammation. Adipocytokines, which are pro-inflammatory cytokines secreted in excess, are elevated in many chronic metabolic diseases. Cytokines and inflammatory mediators, which are not directly linked to DNA, are important in tumorigenesis. Cachexia, a type of metabolic syndrome linked to the disease, is associated with a dysregulation of metabolic pathways. Obesity and cachexia have distinct metabolic characteristics, such as insulin resistance, increased lipolysis, elevated free fatty acids (FFA), and ceramide levels, which are discussed in this section. The goal of this research project is to create a framework for bringing together our knowledge of inflammation-mediated insulin resistance.

## 1. Introduction

Inflammation, metabolic disease, and malnutrition have all recently been studied in scientific research [1]. Oxidative stress causes an increase in the production of highly reactive molecular species in the presence of severe inflammation [2]. Obesity-related metabolic diseases such as insulin resistance, type 2 diabetes, metabolic syndrome, and cardiovascular disease are exacerbated by chronic inflammation and oxidative stress [3,4,5,6]. Nutritional deficiencies have a significant impact on their behavior [7]. Obesity and insulin resistance cause abnormal adipokine, pro-inflammatory cytokine, and acute-phase protein synthesis, all of which contribute to a systemic inflammatory microenvironment. These proteins are found in a variety of organs and are involved in the regulation of metabolic homeostasis [8,9]. This has been linked to the progression of the metabolic syndrome as well as the onset of type 2 diabetes. Nutritional strategies and micronutrient intake can help prevent the severity of oxidative stress and inflammation, thereby aiding in the management and cure of metabolic chronic diseases such as diabetes and obesity [10,11].

## 2. Obesity-Related Inflammation

Obesity is caused by an abnormally large amount of adipose tissue, which is caused by a calorie and fat surplus [12]. Obesity is on the rise throughout the world, and it has been linked to a variety of health issues, including cardiovascular disease, impaired glucose tolerance, and even certain types of cancer [13,14]. Type 2 diabetes is distinguished by insulin resistance, which affects lipid and glucose metabolism in fat tissue, the liver, and the musculoskeletal system [5,15]. Insulin inhibits signaling in the Insulin-Receptor-Substrate (IRS)/PI3K/PKB pathway, which is currently considered to be the most important cause of insulin’s biochemical processes [16,17]. Obesity is now linked to the development of insulin resistance as a result of chronic low-grade systemic and local systemic inflammatory processes that occur throughout the course of obesity [18,19]. This inflammatory condition has been shown to affect fatty tissue, the liver, the endocrine pancreas, the hypothalamus, and possibly the musculoskeletal system. Metabolic inflammation, also known as metainflammation, is a type of chronic inflammation caused by a nutritional excess in the diet [20,21]. Metabolic inflammation is thought to be caused by a variety of factors, including changes in the gut microbiome and lipid and glucose metabolism, which cause cells involved in metabolic regulation and immunity to malfunction [22,23,24]. Obese people’s adipose tissue hypoxia has been linked to adipose tissue hyperinflammation and insulin resistance [25]. Excess nutrients are linked to the production of inflammatory cytokines via pattern recognition receptors (PRRs), one of the mediators [26,27]. Inflammation mediators activate intracellular signal pathways, which influence insulin signal desensitization [28,29].

## 3. Obesity-Induced Inflammation Is Primarily Caused by Specific Recognition Receptors

Obesity increases the production of lipopolysaccharides (LPS), which are produced by commensal Gram-negative bacteria, and activate Toll-like receptors-4 (TLR4). TLR2 and Nod1/2 (Nucleotide Oligomerization Domain) can be activated by peptididoglycans, lipoproteins, and lipopolysaccharide from Gram-positive and -negative meals [26,30,31]. Saturated fatty acids and their metabolites, ceramides, can also interact with TLR4 or indirectly activate it through the TLR4 interface by producing DAMPs such as HMGB1 [27,32]. Saturated fatty acid receptors, such as TLR2, may also exist. Chemokines such as IL-1 and IL-18 are produced after these receptors are activated [33,34]. The mature versions of these molecules should be handled by the inflammasome (which is responsible for triggering inflammatory processes) (NLRP3, ASC, and caspase-1) [35,36]. Caspase-1 is activated in the inflammasome complex when the interaction between TXNIP and NLRP3 is activated by large amounts of reactive oxygen species (ROS) derived from fatty acids, ceramides, or glucose [37,38], all of which are immune sensors that link dietary stress to obesity-related inflammation and insulin resistance.

## 4. Toll-Like Receptors (TLRs)

TLRs are members of the recognition receptor-PRR family and are important for innate immunity because they can recognize pathogen-specific genetic features as well as detect tissue damage [39,40]. In the presence of obesity, TLR2 and TLR4 play a role in the production of inflammation and insulin resistance [41,42]. The primary TLR4-expressing cells are macrophages, dendritic cells, and adipose tissue [43,44]. Obese mice and diabetic people have higher levels of TLR4 expression, which correlates negatively with insulin sensitivity [45]. By activating TLR4 in metabolic tissues during obesity, metabolic endotoxemia is thought to play a role in the development of inflammation and metabolic disorder [46,47]. Metabolic endotoxemia is caused by an increase in plasma lipopolysaccharides (LPS) due to lipopolysaccharide production by Gram-negative bacteria. To understand why this occurs, we must consider changes in the microbiota of the digestive tract, as well as an increase in intestinal permeability [48,49]. High-fat diets have been shown to promote Gram-negative bacteria migration from the gut to fatty tissue, a mechanism that is partially dependent on CD14 (a macrophage-expressed gene), which functions as a co-receptor for LPS recognition alongside TLR4 [50,51]. As a result, it has been demonstrated that the intestine microbiota is to blame for the development of insulin resistance [52,53]. Complex interactions between the host’s genes and the environment, including bacteria in the gut, determine the metabolic phenotype. In adipose tissues and macrophages, saturated fats can act as TLR4 ligands, causing the release of inflammatory mediators such as cytokines and ceramides, which can cause injury [54]. When adipocytes and macrophages detect fatty acids, TLR4 allows them to communicate with one another. Saturated fatty acids have the potential to interact with TLR4 via fetuin-A [55,56].

## 5. Nucleotide Oligomerization Domain (NOD)

In Gram-negative and Gram-positive bacteria, the intracellular proteins NODs 1 and 2 recognize peptidoglycan portions of the cell wall [57,58]. When inflammation leads to insulin resistance, NOD proteins function as immune system sensors, detecting inflammatory mediators and signaling MAP-kinases that activate IRS1’s desensitization function [59,60]. The ability of Gram-negative bacteria to detect components demonstrates the ability to activate NOD insulinoresistance and supports the role of NOD1 in metabolic disease control [61,62].

## 6. Inflammasome

Inflammasomes are multiprotein structures found within cells that form as a result of exposure to pathogen-associated molecular patterns (PAMPs) or other types of cellular or tissue damage (DAMPs) that can cause an inflammatory response [63]. Inflammasome-activated responses are important not only as an antimicrobial response, but also in metabolic and immunological pathways. Inflammasomes form as a result of an immune response to either external or internal stimuli. Inflammasomes, which are responsible for triggering inflammation in the innate immune system, are made up of three proteins (NLR, ASC, and caspase-1) [64]. Caspase-1 is activated when these multiprotein complexes form, which increases interleukin-1 alpha (IL-1 alpha) and interleukin-18 cytokines [65,66]. The cytokine IL-1 plays an important role in the process of insulin signal desensitization [67]. Obese mice, overweight people, and obese people with type 2 diabetes have all been shown to have increased NLRP3 and caspase-1 expression, among other things [68]. In adipose tissue, fat cells and macrophages express and activate NLRP3 inflammasomes [69]. To summarize, these findings strongly support the idea that the “inflammasome NLRP3” detects danger signals caused by obesity, resulting in increased inflammation and organ failure (Figure 1) [70,71]. At this time, we do not know what causes the NLRP3 inflammasome to activate in obese people. Because of the activation of fatty acids and ceramides, the lipotoxic environment of obesity may activate the inflammasome NLRP3 [72]. Conversely, comparable molecules such as ATP, glucose, bad cholesterol, uric acid, and cholesterol stearate crystals (CSG) can activate the NLRP3 inflammasome [73,74]. Given that all of these are elevated in obesity, more research into their distinct roles in activating this pathway is required. All of these warning signs have the potential to increase ROS production, which is required for NLRP3 inflammasome activation [75,76]. As a result, NLRP3 may be linked to oxidative stress in various metabolic tissues [77]. Pathogen-sensing kinases (PKR), immune system sensors (TLR, NOD, inflammasome), and others have all been implicated in metabolic inflammation. Pathogenic and danger-sensitive pathways may be activated in obese individuals, resulting in immune system activation across the body’s metabolic organs [78,79]. To put it another way, when the body is overfed, micronutrients are perceived as potentially dangerous biomolecules, activating pathways normally reserved for detecting endogenous or pathogenic danger signals. Obesity-related changes in the intestinal microbiota and permeability may provide the body with aggressive chemicals such as LPS and some other microbial pathogens, or may facilitate commensal bacteria translocation from the gut to metabolic organs [80,81]. As a result, inflammatory cytokines are produced more frequently, causing metabolic cells to use different signaling pathways that not only deactivate the insulin signal, causing insulin resistance, but also change the expression of proteins that aid in glucose transport [82,83]. The hyperactivation of the IKKß/NF-kb immunological regulator pathways is also a critical mechanism linking systemic inflammatory response and insulin resistance in peripheral tissues and the central nervous system [84,85].

## 7. Adipocytokines

In recent years, we have made significant progress in understanding the molecular structure of fat. This has given us a better understanding of how our bodies function and how they become out of balance as we gain weight. We know that fat can assist the body in balancing its energy intake and output via both peripheral and central pathways [86,87]. As previously stated, it also plays an important role in host defense. Obesity, on the other hand, causes a slew of negative consequences when a positive energy balance is maintained. It also affects other parts of the body, such as the muscle and liver. Adipocytes, which are fat cells, contribute to chronic low-grade inflammation [88,89,90]. Intestinal bacteria, endotoxins, and LPS, as well as food components and nutrients such as glucose and lipids, control inflammation [11,91]. Increased macrophage infiltration causes inflammation by enlarging adipose tissue deposits and causing adipocyte hypertrophy [92,93]. Adiponectin (which is preferentially released by fat tissue), TGFß, interleukin-10 (IL10), IL4, IL13, ILRa, and apelin are all anti-inflammatory adipokines that fat tissue preferentially secretes in thin people [94,95]. Obesity decreases the production of anti-inflammatory adipokines such as adiponectin while increasing the production of inflammatory-promoting adipocytokines and interleukins such as IL-6, TNF and leptin [96,97]. Adipokines appear to serve a variety of functions in people of all weights, not just the overweight or obese. Insulin resistance can be treated by either directly interfering with the insulin signal pathway or by inducing inflammatory responses in individuals with normal levels of metabolic adipokine disease.

## 8. Neoplastic Cachexia

Increased protein and fat catabolism is linked to higher levels of pro-inflammatory cytokines even in protein-caloric deficiency [98,99]. Weight loss in cancer patients is associated with systemic inflammation, which is also involved in the various stages of carcinogenesis [100]. Cancer Cachexia is a subtle condition that reduces patients’ quality of life by impairing their response to therapy and survival [101,102,103]. Patients with sarcomas and breast cancer have a lower risk of developing cachexia than those with pancreatic and stomach tumors, which affect 80 to 90 percent of patients [104,105,106]. A 10% weight loss is associated with a poor prognosis and a shorter life expectancy [107,108]. Obesity and its associated diseases have contributed to a better understanding of the physiological requirements for metabolic stability and a healthy body weight. Inflammatory mediators that may play a role in the pathogenesis of neoplastic cachexia, for example, overlap with those that may play a role in the pathogenesis of obesity [109,110]. Cachexia is a complication of cancer-related malnutrition associated with catabolic/hypermetabolic changes [111,112]. The role of systemic inflammation has been investigated as a possible explanation for the lack of a link between cancer patients’ nutritional intake and weight loss [113,114]. Cachexia is commonly defined as any clinical situation involving circulating inflammatory markers such as IL-6, hyperinsulinemia, weight loss, and poverty [115,116,117]. A clear distinction was made between cachexia and malnutrition, where weight loss could not be attributed solely to calorie restriction [118]. Malnutrition can occur in cancer patients as a result of tumors that directly affect the gastrointestinal tract, chemotherapy nausea, early satiety, and a decreased ability to regulate appetite [119,120].

## 9. Malnutrition and Cancer

A change in nutritional status is common during the natural history of a neoplastic disease [121,122]. Even with adequate nutritional support, neoplasia has a limited reversibility, which distinguishes malnutrition from “regular” malnourishment [123]. Because of this characteristic, this condition is referred to as neoplastic cachexia rather than malnutrition associated with neoplasia [124]. Cachexia may be caused by a tumor–host interaction [125]. Tumors may produce pro-inflammatory cytokines (TNF-, IL-6, and so on), lipid mobilization factors (LMF), and catabolic-inducing protides (CIP) [126,127,128]. If a tumor is present, the host’s inflammatory and neuroendocrine stress responses will be activated [129,130]. In this case, changes in body composition (and their functional consequences) as well as anomalies in the humoral system would emerge. Neoplastic cachexia is defined by weight loss and metabolic abnormalities across multiple substrates, as follows:Improved glucose metabolism (from lactate and amino acids in the muscles), increased lactic acid synthesis, peripheral insulin resistance, and increased lactate excretion [131,132].Increased protein metabolism is defined by an increase in blood levels of a factor that stimulates proteolysis (PIF), an increase in muscle tissue protein catabolism, and a decrease in lean mass and liver protein synthesis [133,134].LMF (lipid mobilizing factor) is an enzyme that increases lipid metabolism, beta-oxidation, and turnover-free fatty acid synthesis to promote lipolysis [135,136].

## 10. Dysregulation of Metabolic Pathways during Cachexia

The long-term release of tumor cytokines into the bloodstream may disrupt the neuroendocrine control of metabolism in multiple organs [137,138]. If metabolites were difficult to obtain or were used incorrectly, cachexia would worsen [139,140]. As a result of the activation of the brown adipose tissue (BAT), which causes a fever, patients with anorexia and cachexia are committed to energy-intensive, maladaptive reactions to anorexia. Obesity and cachexia are associated with increased lipolysis, elevated free fatty acid (FFA), ceramides, and insulin resistance [141,142]. Other metabolic pathways also changing in an unusual way [143,144,145]. Insulin resistance is important in the context of pathogenic bacteria because it diverts nutrients away from anabolic and antimetabolic pathways, and toward immune system energy supplies. This lays the groundwork for understanding insulin resistance caused by inflammation [146]. Immune cells that have been stimulated can receive nutrients via an energy recourse reaction using this adaptive technique [147]. Once the infection has been cleared and equilibrium has been restored, anti-inflammatory chemicals are typically used to counteract this response [148]. Chronic inflammatory disorders such as inflammatory arthritis and chronic obstructive pulmonary disease (COPD) share molecular recognition sensors and mediators with cancer and obesity [149].

## 11. Cancer Cytokines and Inflammation

Proinflammatory cytokines, which play a role in the development and progression of cancer, play a critical role in survival rates, quality of life, and therapy response [150,151]. Many types of cancer patients have elevated levels of cytokines and soluble receptors in their blood. IL-6 and IL-8 concentrations appear to be strongly predictive of prognosis and outcome in a number of studies [152,153,154,155]. When cancer cells interact with other cells in the tumor microenvironment, such as endothelial and immune cells, as well as necrotic tissue, they can process cytokines [156]. The release of cytokines into the bloodstream by tumors has the potential to affect organs located far from the tumor’s location. In contrast to their well-known normal biological roles, these tumor cytokines have the potential to subvert physiological systems when produced chronically by tumors in the absence of sufficient negative feedback regulatory signals [157,158]. Some of these cancer cytokines may be difficult to identify because they are produced by a diverse mix of malignant and normal cells in the tumors of different patients. Because tumors produce and express them in high quantities, their function differs from those of cytokines and myokines, which are also produced and expressed in high quantities by tumors [159,160]. Tumors produce IL-6 continuously, in contrast to immune cells’ precise circadian regulation of IL-6 and other cytokines, and skeletal muscle’s transitory spikes in plasma IL-6 production during exercise. IL-6 concentrations, on the other hand, can increase up to 100-fold during physical activity but quickly return to pre-exercise levels once the activity is completed [108,161,162]. Cachexia has a negative impact on the health and well-being of cancer patients. Individuals with cachexia’s clinical and nutritional care will be improved if the metabolic anomalies and treatment options are better understood [162]. A tumor’s overproduction of cytokines alters the energy balance in many organ systems and reveals treatment options. Researchers can investigate cancer cytokines’ normal and malignant functions, as well as their interactions with inflammatory signals and metabolic abnormalities, to identify cancer cytokines. [163]. Researchers now have new insights into both obesity and neoplastic cachexia, which share many of the same molecular recognition sensors and process mediators [164,165].

## 12. Concluding Considerations on Chronic Inflammation in Obesity and Cancer Cachexia

Researchers now have a new avenue for studying inflammation and how it affects metabolic pathways after discovering that metabolic illnesses are frequently accompanied by a low-grade inflammatory condition. The primary cause of metabolic homeostasis disruption appears to be intercellular communication between immunological and metabolic cells. When a person overeats, immune sensors such as TLRs or inflammasomes detect high levels of lipid in the diet or their metabolic products, resulting in the production of inflammatory cytokines in various metabolic organs. Furthermore, dietary lipids can alter gut bacteria, resulting in an increase in proinflammatory molecules, which can result in an incorrect immunological response. Saturated fatty acids, inflammatory cytokines, and bacterial lipopolysaccharides, LPS, all influence insulin signaling, altering cellular metabolism. As a result, in addition to the obvious strategy of reducing caloric intake, saturated fats, and a low glycemic index, which has been shown to lower insulin levels and reduce systemic inflammation and relapse, new therapeutic strategies aimed at targeting immune sensors and different protein kinase can also be used to improve obesity-related complications. Despite growing scientific attention, this nutritional problem requires a better understanding and a more precise formal description by the health care industry in order to implement numerous therapies that must be tailored to each patient’s needs. The “diet” (in the broadest and most original sense of “lifestyle”) retains its extraordinary importance and merits careful consideration, according to the most recent scientific evidence, given the positive effect demonstrated in numerous studies of physical activity on the associated symptomatology of cachexia.

## Figures and Tables

**Figure 1 jcm-11-02191-f001:**
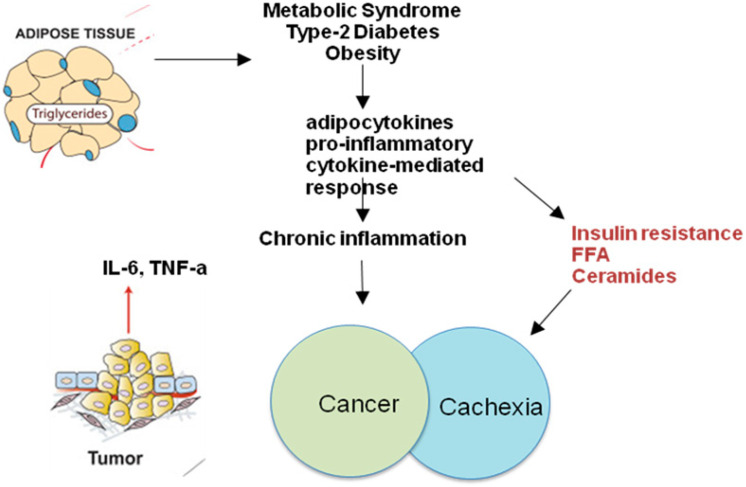
Obesity and related pathologies increase the secretion of proinflammatory adipocytokines by activating an inflammatory response mediated by cytokines. Chronic inflammation and insulin resistance can foster a suitable environment for neoplastic growth and the development of cachexia. Obesity and cancer cachexia, despite the apparent dichotomy, share some common underlying mechanisms that lead to profound metabolic perturbations. Insulin resistance, adipose tissue lipolysis, skeletal muscle atrophy, and systemic inflammation are key players in both diseases.

## Data Availability

The study did not report any data.

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
