# Peer review of "Chronic Inflammation in Obesity and Cancer Cachexia"

_jcm, 2022, doi:10.3390/jcm11082191_

Round 1

Reviewer 1 Report

Main comment:

(1) This manuscript deals with chronic inflammation in individuals with obesity or cancer cachexia. It does not really break new ground, but gives an overview on the subject and on some current developments. It may also be seen as an incentive to further research.

Specific comments/suggestions:

(2) Introduction, line 53: "Metabolic inflammation, also known as metal inflammation" – meta-inflammation?

(3) Line 77/78: "Immune sensors that link stress nutritional to obesity-induced inflammation and insulin resistance Toll-like receptors" – please improve this headline.

(4) Line 102: "peptididoglycan" -> peptidoglycan.

(5) Line 107: "disease control metabolic" – metabolic disease control?

(6) Lines 112-114: "Inflammasome-activated activities are important as an antimicrobial response, but so are normal metabolic pathways and immunological responses" – please clarify this statement.

(7) Line 147: "ikkß/NF-kb" -> IKKß/NF-κB.

(8) Figure 1: Please provide a caption.

(9) Lines 165-167: "The production of pro-inflammatory adipocytokines such as TNFa, IL6, leptin, visfatin, and resistin, as well as interleukins, increases in obese persons, whereas the production of anti-inflammatory adipokines decreases" – as IL6 is an interleukin, "IL6 […] as well as interleukins" should be rephrased.

(10) Line 228: "via a energy recourse reaction" -> via an energy recourse reaction.

(11) The author contributions should be specified.

(12) References: Reference 171 is not cited within the text. Number "172" is to be deleted in the reference list.

Author Response

Main comment: first reviewer

(1) This manuscript deals with chronic inflammation in individuals with obesity or cancer cachexia. It does not really break new ground, but gives an overview on the subject and on some current developments. It may also be seen as an incentive to further research.

Specific comments/suggestions:

Answer (2) Introduction, line 53: "Metabolic inflammation, also known as metal inflammation" – meta-inflammation?

Reply: regarding point two we have corrected the phrase metal inflammation in meta inflammation.

Answer (3) Line 77/78: "Immune sensors that link stress nutritional to obesity-induced inflammation and insulin resistance Toll-like receptors" – please improve this headline.

Reply: as regards point 3 we have reformulated the sentence by improving the form: Immune sensors that link dietary stress to obesity-related inflammation and insulin resistance.

Answer (4) Line 102: "peptididoglycan" -> peptidoglycan.

Reply: we corrected the term with peptidoglycan

Answer (5) Line 107: "disease control metabolic" – metabolic disease control?

Reply: we corrected the sentence metabolic disease control

Answer (6) Lines 112-114: "Inflammasome-activated activities are important as an antimicrobial response, but so are normal metabolic pathways and immunological responses" – please clarify this statement.

Reply: as regards point 6 we have reformulated the sentence by improving the form: Inflammasome-activated responses are important not only as an antimicrobial response, but also in metabolic and immunological pathways.

Answer (7) Line 147: "ikkß/NF-kb" -> IKKß/NF-κB.

Reply: we corrected as suggested by the reviewer: IKKß/NF-κB

Answer (8) Figure 1: Please provide a caption.

Reply: for this point we have proceeded to insert a description of figure 1

Answer (9) Lines 165-167: "The production of pro-inflammatory adipocytokines such as TNFa, IL6, leptin, visfatin, and resistin, as well as interleukins, increases in obese persons, whereas the production of anti-inflammatory adipokines decreases" – as IL6 is an interleukin, "IL6 […] as well as interleukins" should be rephrased.

Reply: we rephrased the sentence following the reviewer's suggestion: Obesity decreases the production of anti-inflammatory adipokines like adiponectin while increasing the production of inflammatory-promoting adipocytokines and interleukins like IL-6, TNF and leptin

Answer (10) Line 228: "via a energy recourse reaction" -> via an energy recourse reaction.

Reply: the sentence was reworded as suggested by the reviewer

(11) The author contributions should be specified.

Answer (12) References: Reference 171 is not cited within the text. Number "172" is to be deleted in the reference list.

Reply: reference 171 has been added in the text and reference 172 has been deleted from the bibliography section.

Reviewer 2 Report

In the present review, article authors have summarised the role of chronic inflammation in obesity and cancer cachexia. Authors have discussed obesity-related inflammation, the role of adipocytokines, and cancer-associated cachexia. Information presented is very important, but the connection between the different chronic conditions and their interconnect is lacking. Similarly, molecular aspects of different cytokines-induced wasting are lacking. It will be great if authors can present some more illustrations depicting the interconnection between these conditions and the molecular mechanism of wasting.

Author Response

Second reviewer

In the present review, article authors have summarised the role of chronic inflammation in obesity and cancer cachexia. Authors have discussed obesity-related inflammation, the role of adipocytokines, and cancer-associated cachexia. Information presented is very important, but the connection between the different chronic conditions and their interconnect is lacking. Similarly, molecular aspects of different cytokines-induced wasting are lacking. It will be great if authors can present some more illustrations depicting the interconnection between these conditions and the molecular mechanism of wasting.

Reply: we have described the connections of these chronic states in figure 1